# Learning How Hard to Think: Input-Adaptive Allocation of LM Computation

**Mehul Damani**[*]  **Idan Shenfeld**  **Andi Peng**  **Andreea Bobu**  **Jacob Andreas**
Massachusetts Institute of Technology

## Abstract

Computationally intensive decoding procedures—including search, reranking, and self-critique—can improve the quality of language model (LM) outputs in problems spanning code generation, numerical reasoning, and dialog. Existing work typically applies the same decoding procedure for every input to an LM. But not all inputs require the same amount of computation to process. Can we allocate decoding computation *adaptively*, using more resources to answer questions whose answers will be harder to compute? We present an approach that predicts the distribution of rewards given an input and computation budget, then allocates additional computation to inputs for which it is predicted to be most useful. We apply this approach in two decoding procedures: first, an *adaptive best-of-k* procedure that dynamically selects the number of samples to generate as input to a reranker; second, a *routing* procedure that dynamically responds to a query using a decoding procedure that is expensive but accurate, or one that is cheaper but less capable. Across a suite of programming, mathematics, and dialog tasks, we show that accurate computation-allocation procedures can be learned, and reduce computation by up to 50% at no cost to response quality, or improve quality by up to 10% at a fixed computational budget.

## 1 Introduction

Recent improvements in language models (LMs) have dramatically enhanced their ability to tackle complex tasks in mathematics, coding, and reasoning. However, as with natural and artificial intelligent agents of all kinds (Silver et al., 2016), LMs cannot solve all problems on the first try: they benefit from the ability to perform search (Yao et al., 2024), sampling (Brown et al., 2024), or more sophisticated decoding procedures like chain-of-thought (Wei et al., 2022) and self-critique (Wang et al., 2023).

Importantly, computationally intensive problem *domains* may exhibit considerable variation in the difficulty of individual problem *instances*: not all problems are equally hard to solve. For example, even a novice programmer can likely write code to to test if an integer is even. Balancing a binary tree might require multiple attempts, and finding a polynomial-

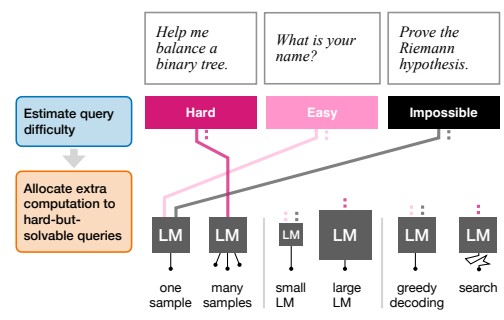

Figure 1: Overview of our approach. Given a set of input queries to a language model, we train a lightweight model to estimate the difficulty of these queries (more precisely, a model that estimates how much each query would benefit from a more computationally intensive decoding procedure.) We then allocate extra computation to those queries for which it would be most beneficial.

time set cover algorithm is likely impossible, and not be worth attempting at all. Maximally efficient use of computational resources thus requires identifying, *a priori*, the inputs for which additional computation will improve outputs. In LMs, recent work has shown that significant gains from adaptive choice of decoding procedures are theoretically achievable (Snell et al., 2024). However, past

---

[*]Correspondence to `mehul42@mit.edu`.

work has relied on measures of problem difficulty that could only be estimated *a posteriori*, using a sampling procedure significantly more expensive than the one used to generate model outputs.

In this paper, we show that it is possible to perform inference-time scaling of computation *before* responding to a set of user queries, by predicting how much computation will be required to respond successfully to each query, and allocating resources accordingly (Fig. 1). Our approach has two parts. First, we describe how to train a *difficulty model* that estimates the marginal improvement in response quality that would result from allocating an additional unit of computing power when responding to a query. We show that effective difficulty models can be constructed by training lightweight "probes" on top of a pre-trained LM's hidden representations, suggesting that LMs may already learn to encode problem difficulty as a result of pre-training. Second, we describe an efficient allocation algorithm that, given a collection of queries and a computation budget, uses predictions from the difficulty model to allocate computation to individual queries. We show how to perform this optimization online (exactly satisfying the budget for a batch of queries) or offline (computing a fixed mapping from queries to compute allocations so that constraints are satisfied in expectation).

This approach is flexible with regard to the choice of compute-scaling procedure, and we demonstrate its effectiveness with two different inference procedures. In **best-of-*k*** experiments, we choose how many samples to generate from an LM before re-ranking with a reward model. In **routing** experiments, we choose whether to respond to queries using a computationally expensive but capable decoding procedure, or a less powerful but more cost-effective decoder. For the best-of-$k$ setting, we present results across 3 domains: Math, Code and Chat. Adaptive sampling outperforms non-adaptive strategies on all three domains across a range of possible compute budgets. On Math and Code, we achieve the same performance as non-adaptive methods using up to 50% less compute; on Chat, we are able to match reward using up to 10% less compute than base models. For routing, we experiment with both a pair of models (Gemma-2B, Gemma-7B) and a pair of decoding procedures (ordinary sampling and value-augmented sampling; Han et al., 2024). In both cases, we match the performance of the more expensive decoder while calling it only 50–75% of the time.

In summary, this work presents (1) a generally applicable framework for adaptively scaling test-time LM computation; (2) a learned model for estimating the marginal benefit of additional computation in LM decoding; (3) an efficient algorithm for allocating computation to queries; and (4) experiments demonstrating significant improvements in LM efficiency and output quality.

## 2 PRELIMINARIES

Suppose we have an LM-based agent that will interact with a large number of users in parallel. At any moment, each of a set of users has issued a **query** $x_i$, for which we wish to produce a **response** $y_i$. We have acccess to both a **language model (LM)** $p(y \mid x)$, capable of generating candidate responses, as well as a **reward model** $r(x, y)$ capable of assessing the quality of candidate responses. In the absence of any computational constraints, we might wish to find the best response to every user query by optimizing $\arg\max_y r(x_i, y)$ independently for each query $x_i$.

In practice, however, we generally do not have the ability to perform exhaustive search over candidate responses $y$. Instead, we use a **decoding procedure** $f(x, b)$ that (stochastically) generates a response $y$ subject to some constrained **computation budget** $b$. (In general, increasing the budget to a query should increase the quality of the response.) Many such decoding procedures are in wide use; our experiments will focus on two of the most widely used.

In **best-of-*k***, we generate a finite number of samples, then rerank them:

$$f(x, b) = \underset{y_i \in \{y_1, \ldots, y_b\} \sim p(\cdot|x)}{\arg\max} r(x, y_i) \, . \tag{1}$$

In **routing**, we have access to a **strong** but expensive decoding scheme $p^S$ (which could involve search, reasoning, or even just a larger model). For every query we can either use this decoding scheme or to fall back to a **weak** but cheaper decoding scheme $p^W$ (e.g. ordinary decoding or a smaller model). Then:

$$f(x, b) = \begin{cases} y \sim p^W(\cdot \mid x) & \text{if } b = b^W \\ y \sim p^S(\cdot \mid x) & \text{if } b = b^S \end{cases} \, . \tag{2}$$

Here $b^W$ and $b^S$ denote the cost of calling the weak and strong decoders respectively. Decoding procedures including consensus (Jacob et al., 2024), chain-of-thought (Wei et al., 2022), self-critique (Luo et al., 2024) and debate (Du et al., 2024) may all be expressed in this form, with the budget $b$ controlling the number of generated samples, tokens, or rounds of revision.

In current practice, it is typical to set a *fixed* budget $B$, and allocate this uniformly for all queries—returning $y_i \sim f(x_i, B)$ for each $x_i$. In this case, the expected reward for a set of queries is simply:

$$\sum_i \mathbb{E}_{y_i \sim f(x_i, B)}[r(x_i, y_i)] \tag{3}$$

But as noted in Section 1, different $x_i$ may in practice benefit differentially from increased budgets. In this paper, we consider a more flexible version of the problem—if we are willing to set a fixed *average* computational cost per query, can we improve overall accuracy by allocating this computation adaptively? In this case, responding to a collection of queries $\{x_1, \ldots, x_n\}$ corresponds to solving:

$$\max_{b_1, \ldots, b_n} \sum_i \mathbb{E}_{y_i \sim f(x_i, b_i)}[r(x_i, y_i)] \quad \text{s.t.} \quad \sum_i b_i \leq B \cdot n \tag{4}$$

Below, we develop a method for solving this **adaptive comptuation scaling problem** efficiently.

## 3 METHOD

In Eq. (4), we would like to allocate computation $b_i$ to queries $x_i$ to maximize **expected reward**, which we will denote $q(x_i, b_i) = \mathbb{E}_{y_i \sim f(x_i, b_i)}[r(x_i, y_i)]$. But there is a problem: without querying the LM, we cannot know the value of $q(x_i, b_i)$ for some new $x_i$. To make matters worse, reliable Monte Carlo estimation of $q(x_i, b_i)$ (as in Snell et al., 2024) may require more computation (e.g. more LM samples) than we eventually wish to allocate to $x_i$! To efficiently allocate computation, we need some way to *predict* $q(x_i, b_i)$ from $x_i$ alone, and then use these predictions to solve Eq. (4).

When estimating problem difficulty and allocating decoding budget, it will be useful to reason about the benefits of *incremental* changes in compute budgets. Formally, let us define the **marginal reward** $\Delta_{ij} = q(x_i, j) - q(x_i, j-1)$ (with $\Delta(\cdot, 0) = 0$). Under this definition, $q(x_i, b_i) = \sum_{j=1}^{b_i} \Delta_{ij}$; intuitively, $\Delta_{ij}$ represents the expected gain from allocating one more "unit" of computation to $x_i$, given that we have already allocated $j - 1$ units. We may then re-write Eq. (4) as:

$$\max \quad \sum_{i=1}^n \sum_{j=0}^{B_{\max}} c_{ij} \Delta_{ij} \quad \text{s.t.} \quad \sum_{i,j} c_{ij} \leq B \cdot n \, ; \quad c_{ij} \leq c_{i,j-1} \, \forall i, j \tag{5}$$

Here we have introduced auxiliary variables $c_{ij} \in \{0, 1\}$ to represent budget increments: intuitively, setting $c_{ij} = 1$ means we have allocated a $j$th unit of computation to $x_i$, and constraints enforce that these units are allocated in sequence, as shown in Figure 2. To determine how to allocate decoding computation across a set of queries $x_i$, it then suffices to predict the value of each term $\Delta_{ij}$ and then solve Eq. (5) to obtain $c$. Each of these steps is described in more detail below.

### 3.1 ESTIMATING PROBLEM DIFFICULTY BY PREDICTING MARGINAL REWARDS

Given a training set of queries $x_i$, we collect empirical estimates of $\Delta_{ij}$ by decoding from $f(x, b)$ at values of $b$ up to some $B_{\max}$. We then train a model $\hat{\boldsymbol{\Delta}}(x_i; \theta)$ that predicts a vector of marginal rewards for all budgets $j$ simultaneously by optimizing the mean squared error:

$$\arg\min_{\theta} \sum_{x_i, \Delta_i} \|\boldsymbol{\Delta}_i - \hat{\boldsymbol{\Delta}}(x_i; \theta)\|_2^2 \tag{6}$$

where $\boldsymbol{\Delta}_i = [\Delta_{i1}, \ldots, \Delta_{iB_{\max}}]$ is the vector of empirical marginal rewards for the query $x_i$. While training this model requires sampling from the LM at many budgets to obtain supervision, it can be called during inference without generating any outputs at all.

We explore two simple parameterizations of $\hat{\boldsymbol{\Delta}}$: (1) a two-layer MLP that takes last the hidden state of the base LM $p$ (obtained by encoding the query) as input, and (2) parameter-efficient fine-tuning of the base LM with LoRA (Hu et al., 2022). During inference, the MLP variant adds extremely little overhead as its input are hidden states that are already computed as part of decoding procedure. The LoRA variant is slightly more expensive; however, its overhead is also negligible, as the primary bottleneck during inference occurs when decoding outputs rather than encoding queries.

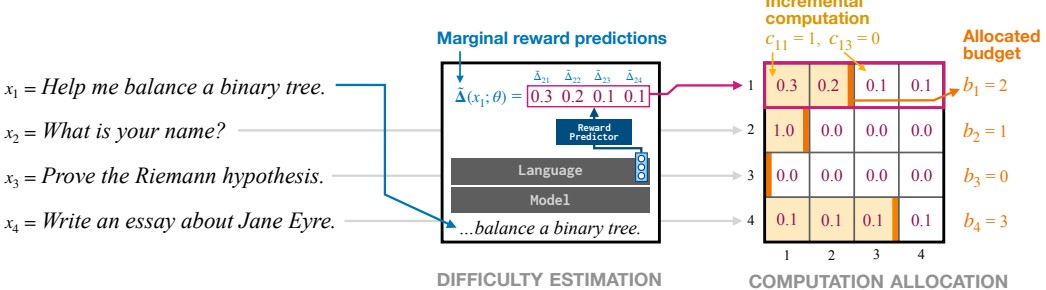

Figure 2: **Method overview**. In the first step, the marginal reward predictor is used to estimate the marginal reward gains for a batch of queries. In the second step, the allocation algorithm uses these predictions to assign compute budgets to individual queries. The simple query *what is your name?* is allocated a budget of 1, while the harder query *balance a binary tree* is allocated a budget of 2.

## 3.2 ALLOCATING COMPUTATION

Eq. (5) is an integer linear program. But under the assumption of *decreasing* marginal benefits from additional computation, it has a special form that allows it to be solved in $O(B_{\max} \cdot n)$ time using a greedy algorithm that incrementally "turns on" $c_{ij}$ for which $\Delta_{ij}$ is largest. We exploit this property in two ways:

**Online allocation:** If queries $\{x_i\}$ are known *a priori*, we simply replace each $\Delta_{ij}$ in Eq. (5) with the corresponding estimate from $\hat{\Delta}(x_i; \theta)$, solve for $c_{ij}$, then finally set each $b_i = \max_j c_{ij}$.

One drawback of this procedure is that responses must be processed in a batch. However, in some cases it is also possible to effectively set budgets independently for each $x_i$:

**Offline allocation:** If allocations must be made without access to a full batch of samples, we construct a fixed allocation policy as follows:

**(1)** Hold out a subset of the data used for training the reward estimator $\hat{\Delta}$, then use it to label queries in this held-out set (e.g. based on the first-sample prediction $\hat{\Delta}(x_i)_1$). Divide these queries into a fixed set of bins according to their predicted marginal rewards.

**(2)** Solve the allocation problem for this held-out as in Eq. (5), with the additional constraint that all queries in a bin receive the same budget allocation. For each bin, store the assigned budget.

**(3)** During deployment, compute a reward prediction for each $x_i$, map it to a bin, and return the budget associated with that bin.

During deployment, all queries can be processed independently (at the risk of a budget violation if the distribution of queries is significantly different from those used to compute the allocation policy).

## 3.3 INTERESTING SPECIAL CASES

**Binary Reward + Best-of-$k$:** In domains such as **coding** and **math**, rewards are often binary, indicating success or failure. For instance, an outcome reward model can be used to assess correctness in math while unit tests indicate correctness in coding. In such settings, the probability that a single sample from the model succeeds may be used to analytically compute all marginal rewards.

Let $\lambda = \mathbb{E}_{y \sim p(\cdot|x)}[r(x, y)]$ denote the probability of obtaining a successful result from a single sample (for $r(x, y) \in \{0, 1\}$). Then the reward estimate $q(x, b)$ is simply the probability of getting at least one success in $b$ attempts: $q(x, b) = 1 - (1 - \lambda)^b$. Then $\Delta(x_i, b_i) = \lambda_i(1 - \lambda_i)^{b_i}$.

In this case, rather than training the reward predictor $\hat{\Delta}$ using a squared loss as in Eq. (6), we obtain empirical estimates of $\lambda_i$ at training, then minimize the cross-entropy:

$$\sum_{x_i, \lambda_i} \left[ \lambda_i \log(\hat{\lambda}(x_i; \theta)) + (1 - \lambda_i)(\log(1 - \hat{\lambda}(x_i; \theta)) \right] \tag{7}$$

with an appropriately parameterized $\hat{\lambda}(x; \theta)$.

**Routing:** For the routing setting, we learn $\hat{\boldsymbol{\Delta}}$ in a special form that models the probability of outputs from the strong model $p^S$ being preferred over the weak model $p^W$ as:

$$\hat{\Delta}(x_i; \theta)_{b^S} \approx p(p^S \succ p^W \mid x) = \mathbb{E}_{y_1 \sim p^S, y_2 \sim p^W}[\sigma(r(x, y_1) - r(x, y_2))] \tag{8}$$

## 4 EXPERIMENTS

We apply our method to several adaptive decoding procedures: best-of-$k$ and routing (to either a large model or a sophisticated search algorithm). We evaluate improvements over standard decoding and procedures that allocate computation uniformly across problem instances, as well as performance relative to the theoretical upper bound. In addition, we provide *intrinsic* evaluations of the accuracy and calibration of reward predictors $\hat{\boldsymbol{\Delta}}$.

### 4.1 ADAPTIVE BEST-OF-$k$

We use best-of-$k$ reranking with a reward model in three settings: Math, Code and Chat.

**Methods:** We evaluate the following methods:

1. **Online Ada-BoK (ours):** The online variant of our method that solves a joint optimization problem, as detailed in Section 3.2.
2. **Offline Ada-BoK (ours):** The offline variant that solves the allocation problem on a held-out dataset, as detailed in Section 3.2. This method is only used in Math and Code domains.
3. **Best-of-$k$:** This baseline allocates the same number of samples $k = B$ to every query.
4. **Oracle:** Oracle is a non-implementable method that uses the ground truth marginal rewards to solve the allocation problem. This method solves the allocation problem by plugging in the true marginal rewards. It is **unrealizable** in practice, but provides an upper bound on the reward that could be obtained if the learned marginal reward predictor $\hat{\boldsymbol{\Delta}}$ were perfect.

**Evaluation Metrics:** Our main evaluation metric for Math and Code is the expected success rate under an oracle verification procedure:

$$\textbf{Expected Success Rate} = \frac{1}{n}\sum_{i=1}^{n}\mathbb{E}_{y_i \sim f(x_i, b_i)}[\mathbb{1}\{y_i \text{ is correct}\}] \tag{9}$$

where $n$ is the number of queries. Similarly, for Chat, it is the expected reward:

$$\textbf{Expected Reward} = \frac{1}{n}\sum_{i=1}^{n}\mathbb{E}_{y_i \sim f(x_i, b_i)}[r(x, y_i)] \tag{10}$$

To estimate these in practice, we sample a large number of generations $B_{max}$ for each query and then use bootstrapping to approximate the expectation for different $b_i$.

The compute budget $B$ is the average number of samples that may be drawn per query. Best-of-$k$ with $k = B$ uses the same number of total samples as our adaptive method. We plot the performance of different methods for different compute budgets, where budgets are specified in terms of $B$.

Note that, in the binary reward-based Math and Code domains, it may be efficient to set $b_i = 0$ for some queries, as many problems have a 0% success rate. In these cases, a default response such as *I don't know* may be returned, allowing the sample budget to be allocated more effectively elsewhere. For chat experiments, we require all $b_i \geq 1$.

### CODE

**Setup:** For the coding experiments, we adopt a subset of **TACO**, a dataset focused on algorithmic code generation with problems sourced from various programming contest platforms (Li et al., 2023). We use a custom off-the-shelf **Starcoder-15B** model, which was released by the creators of the TACO dataset after fine-tuning on the dataset. We use the official open-source evaluation framework released with TACO. A generation is classified as a success if it passes all test cases. Finally,

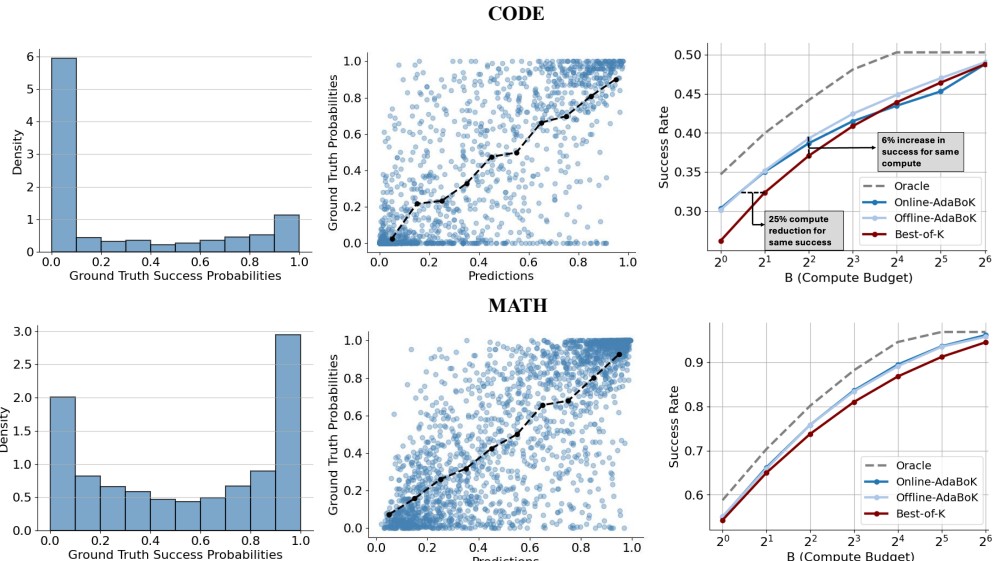

Figure 3: **Results on Code and Math**. The left column shows the distribution of LM success probabilities across queries. A significant number of problems on Code have $0\%$ success rate, while Math has a flatter distribution of difficulty. The middle column compares predicted marginal reward predictors to ground-truth values, alongside calibration curves (our predictors are well calibrated with the ground truth). Finally, the right column shows the performance curves for tested methods. Here, we observe that adaptive computation generally outperforms the best-of-$k$ baseline.

as unit tests serve as a verifier, a method is considered successful on a query if at least one of its $k$ generations passes all test cases. We use the MLP variant to learn the marginal rewards.

**Results:** Figure 3 present success rates for the code dataset. We first observe that while our methods perform significantly better in the low-budget regime ($B < 8$), the results in the moderate-to-high budget regime are mixed. Counter-intuitively, while our offline variant always outperforms best-of-$k$, our online variant actually falls below the best-of-$k$ curve in the high-budget regime. This is because small errors in the learned marginal reward predictor can hugely impact solutions to the allocation problem (if a problem has a true success rate of 0%, but its success rate is predicted to be 1%, it becomes an attractive candidate for a large budget). 50% of problems in the coding dataset have 0 success probability (for Math, this is only 5%). The offline variant avoids this pathology by binning impossible and low-probability solutions together, effectively "regularizing" allocations.

In summary, the coding results highlight the fact that prediction errors in marginal reward can adversely impact online allocation. At the same time, the consistent outperformance of the offline variant over best-of-$k$ underscores the value of adaptive compute budget allocation.

MATH

**Setup:** We use a subset of the **Numina-COT Dataset** (Li et al., 2024), which contains Math problems obtained from diverse sources. We use Mistral AI's **Mathstral-7B** model, which is specialized for mathematical tasks, and is based on Mistral 7B (Jiang et al., 2023). We use this model off-the-shelf and do not perform any fine-tuning. We use an oracle verifier to select the best answer out of $k$. This implies that if the model generates at least one correct answer out of $k$, it will be successful. The verifier uses a 2-stage pipeline that employs the evaluation framework of Hendrycks et al. in the first stage and a custom LLM-verifier in the second stage (Hendrycks et al., 2021). This pipeline is detailed in Appendix A. We used the LoRA variant to learn the marginal rewards.

**Results:** Figure 3 illustrates the success rate across different compute budgets. Our online and offline methods consistently outperform best-of-$k$ for all compute budgets. While the improvements are marginal in the low-budget regime ($B < 8$), adaptive computation shines in the moderate-to-high

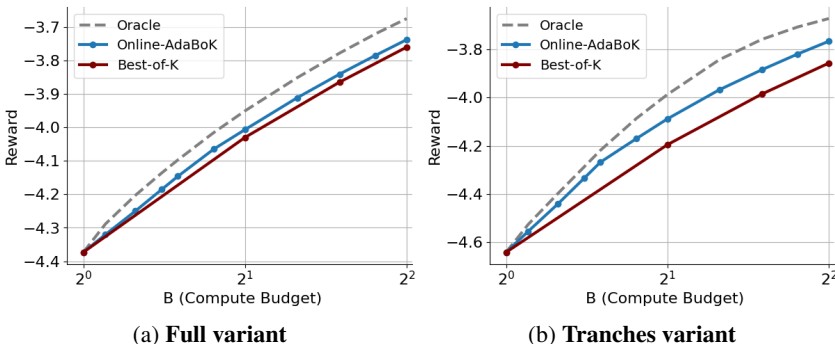

(a) **Full variant**  (b) **Tranches variant**

Figure 4: **Results of adaptive best-of-$k$ in the Chat domain**. Our **full** variant achieves marginal improvements, enabling a 0-10% reduction in compute budget while maintaining equivalent reward levels. More notably, the **tranches** variant exhibits substantial gains, achieving the same reward levels with a 25-40% reduction in compute budget.

budget settings ($B \geq 8$), where they can achieve the same success rate as best-of-$k$ while using 25–50% less computation. This efficiency likely stems from allocation of most of the compute resources to medium and hard problems, which require more samples, while easy problems need only a few to be solved. Both the online and offline variants perform nearly identically, indicating that solving the optimization problem offline does not lead to any performance degradation in this setting. Finally, although we significantly outperform best-of-$k$, the oracle curve suggests that further improvements in the marginal reward predictor $\hat{\Delta}$ could provide even greater gains.

CHAT

**Setup:** For chat, we use a subset of the LMSYS-Chat dataset, which contains real-world conversations with 25 LLMs (Zheng et al., 2024). We use the popular **Gemma-7b-it** model, which is an instruction-tuned model for chat (Team et al., 2024). We use reward as our primary evaluation metric. We use an off-the-shelf reward model called **NCSOFT/Llama-3-OffsetBias-RM-8B**, which was ranked amongst the top 10 on RewardBench at the time of writing. We used the MLP variant to learn the marginal rewards. Finally, we conduct evaluation on 2 different subsets of the dataset-

1. **Full:** The vanilla experiment uses the the entire test set.
2. **Tranches:** The tranches experiment uses a subset of $20\%$ from the full test set. This subset is composed of queries that fall in the lowest $10\%$ or highest $10\%$ of variance in rewards. The goal of the tranches experiment is to simulate a more extreme user distribution. The main goal of the tranches is to simulate a more extreme user distribution compared to typical datasets, which are often collected in somewhat controlled settings and might not capture true user diversity. Full details in Appendix A.3.

**Results:** Figures 4a and 4b present the results for the full and tranches variant respectively. We focus on the small compute budget regime as chat requires much lesser search and we empirically observed rewards saturating quickly. In the case of the *full* variant, the gains are relatively modest. The Oracle curve shows a 15–25% reduction in compute budget while maintaining the same average reward. Although our adaptive method consistently outperforms best-of-k everywhere, the reductions in compute budget are relatively small, ranging from 0–10%. This indicates that while adaptive allocation can provide some benefit, an equitable allocation performs almost at par. For the *tranches* variant, the results are notably different. Our adaptive method achieves substantial gains, reducing the budget by 25–40% while matching the rewards of best-of-$k$.

## 4.2 ROUTING

Through this set of experiments, we aim to evaluate the effectiveness of our learned preference predictor in routing queries between a weak decoding procedure $p^W$ and a strong procedure $p^S$.

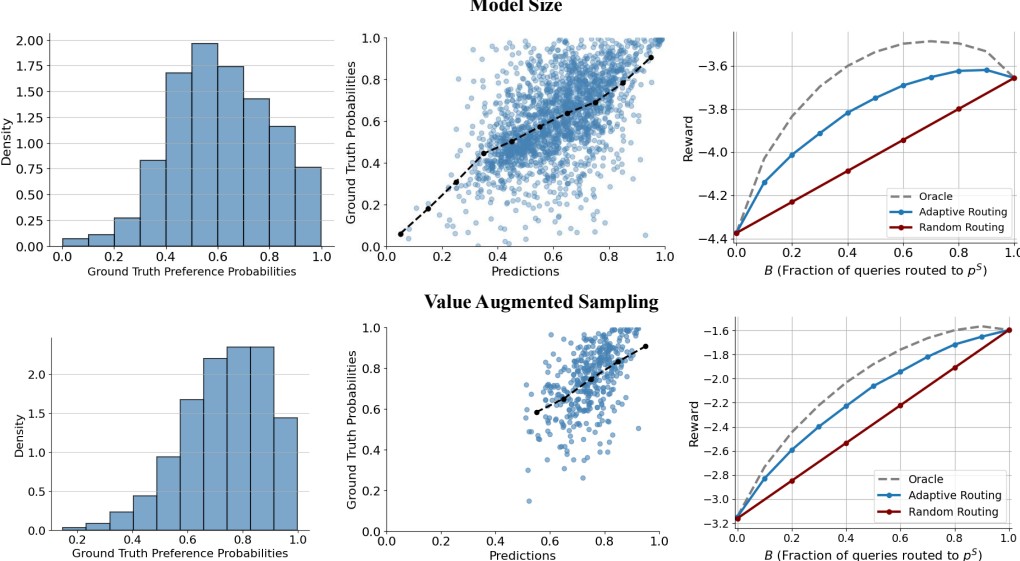

Figure 5: **Results for different routing procedures**. The left column shows the distribution of ground-truth preference probabilities for model size and value-augmented sampling. The middle column compares our learned preference predictors to ground-truth values, alongside calibration curves (our predictors are well calibrated). Finally, the right column shows the performance curves for tested methods. We observe that adaptive routing outperforms the random routing baseline, demonstrating that our learned predictor is able to effectively route more challenging queries to the stronger model while leaving simpler queries to the weaker model.

**Settings:** We present results for two different ($p^W$, $p^S$) pairs which are based on:

1. **Model Size:** In this setting, $p^W$ and $p^S$ are models from the same family but with different model sizes. We use the instruction-tuned **Gemma-2b-it** and **Gemma-7b-it** models as $p^W$ and $p^S$. We use the *full* variant of the LMSYS dataset described in Section 4.1 .

2. **Value-Augmented Sampling:** In this setting, $p^W$ and $p^S$ have the same base LLM model. During decoding, a value function is used to guide search, improving performance but with significant computational overhead. In this experiment we use **Llama-2 7B** for both the LLM and the value function. In each decoding step, we compute the value of 10 possible tokens, increasing the cost of decoding by a factor of 10. We use the harmless subset of the popular Anthropic HH dataset (Bai et al., 2022).

**Methods:** We present results for the following methods:

1. **Online Routing (ours):** $\hat{\Delta}$ is used to predict the preference probabilities for a set of queries. These predictions are then routed using the online allocation procedure.

2. **Random:** A simple baseline that randomly routes a fixed fraction of queries to $\pi^S$. Any target $B$ may be obtained by changing this fraction.

3. **Oracle:** As above, a non-realizable skyline that uses ground-truth information about the reward distribution of $p^W(\cdot \mid x_i)$ and $p^S(\cdot \mid x_i)$ for routing.

**Setup:** We use expected reward, defined in Eq 10, as our main evaluation metric. For reward prediction in experiments with variable model size, we use the smaller $p^W$ as the base LLM. That is, we train using the hidden states of $p^W$ or perform LoRA fine-tuning of $p^W$. This ensures that computational overhead during inference is minimal, and $p^S$ does not even have to be called at all for some queries. The compute budget $B$ is the fraction of the total queries that can be routed to $p^S$. For example, $B = 0.7$ allows routing 70% of the queries to $p^S$.

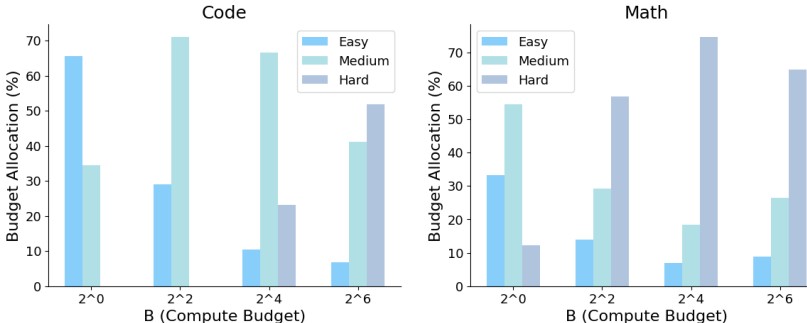

Figure 6: **Allocation of compute at different budgets.** The "Easy", "Medium", and "Hard" categories are obtained by binning queries according to their predicted success probability. In the low-budget regime, the majority of the budget is allocated to easy and medium queries; as budget increases, it is primarily allocated to hard queries.

**Results:** Figure 5 shows that for both experiments, the learned predictor effectively routes the more challenging queries to the stronger model, while simpler queries are handled by the weaker model. This leads to substantial compute savings without a significant loss in overall performance. For example, in Value augmentation, we achieve up to a 25-40% reduction in calls to the more expensive decoding scheme while maintaining similar levels of reward.

Strikingly, for some budgets, our routing scheme actually outperforms the strong decoder—because the weak decoder sometimes outperforms the strong one, routing can achieve better overall performance. Our reward predictor learns and exploits this pattern.

## 4.3 ANALYSIS

### HOW DO THE LEARNED MARGINAL REWARD PREDICTORS PERFORM?

So far, we have seen that our learned marginal reward predictors can be used effectively in adaptive compute allocation. However, we also evaluate their performance independently of the adaptive compute allocation. To this end, we introduce three evaluation metrics:

1. **Avg. (Average Loss):** The empirical loss when the prediction for every query is the average marginal reward, i.e., $\bar{\Delta} = \frac{1}{N} \sum_i \Delta_i$. In this case, the model predicts the same marginal reward for each input. If the representations of the language model do not carry any meaningful information, we expect the performance to approximately equal this average loss.

2. **Opt.\* (Oracle Loss):** The loss that a perfect predictor would achieve. Since we use soft labels rather than binary (0-1) labels, the minimum possible loss is a positive value.

3. **Acc. (Accuracy):** The accuracy of predictions if the median of $\Delta$ values is used as a threshold. That is, if $\Delta_i > median$, the label for that query is 1, and otherwise 0. The accuracy of a random predictor would $50\%$.

Table 1 summarizes the results. In all cases, the achieved loss is lower than the baseline, indicating that the queries contain meaningful signals about the model's response distributions. In routing, the test loss is closer to the baseline, which can be attributed to the low entropy in the reward distribution (Figure 5). Across all settings, accuracy exceeds 70%, showing that difficulty is predictable but highlighting scope for improvement. In Appendices C and D, we test how our predictors generalize to different data distributions and decoding methods.

| Setting | Ours | Avg. | Opt.* | Acc |
|---|---|---|---|---|
| Code | 0.33 | 0.58 | 0.20 | 74% |
| Math | 0.48 | 0.69 | 0.34 | 84% |
| Chat (Model routing) | 0.64 | 0.67 | 0.58 | 72% |
| Chat (VAS routing) | 0.55 | 0.57 | 0.51 | 72% |

Table 1: Learned predictors achieve lower loss than fixed baselines (Avg.), approach optimal values (Min.), and accurately discriminate queries of above- and below-median difficulty (Acc.).

HOW DOES ALLOCATION VARY WITH PREDICTED DIFFICULTY?

Having established the performance benefits of adaptive allocation, we now explore what the distribution of compute budget for different marginal reward predictions looks like. To investigate this, we focus on the Code and Math domains, where allocation is performed using $\lambda_x$, the success probability of a query. We stratify the predicted success probabilities into three **evenly-sized** bins according to their predicted success probability.

Figure 6 illustrates how the compute allocation changes across bins as the budget increases. At the lowest budget, the majority of the allocation goes to queries predicted to be easy or medium in difficulty. At the highest compute budget, by contrast, most of the compute is allocated to the hard bin. This shift can be understood intuitively: easy-to-moderate queries typically require only a few samples to solve, beyond which the marginal gain of additional samples decays rapidly. In contrast, for queries with low success probabilities, the marginal gain remains high even with a large number of samples and decays extremely slowly. Finally, the distinct allocation patterns between the two domains highlight how underlying difficulty distributions significantly impact allocation strategies.

## 5   RELATED WORK

**Decoding Procedures in LLMs.** There has been extensive research into utilizing different decoding schemes to enhance LLM capabilities, usually by expending inference-time compute. One of the most straightforward approaches is best-of-$k$ sampling, where $k$ different model responses are generated per query, and the final model output is selected using a reward model (Gao et al., 2023; Beirami et al., 2024), majority voting (Wang et al., 2023), or verifiers (Li et al., 2022). Another line of work has shown that allowing LLMs to generate intermediate "reasoning" steps, or "chain-of-thoughts" (CoTs) can significantly improve their final answer (Wei et al., 2022; Nye et al., 2021; Zhou et al., 2023). Extending this, some recent works have focused on training models to generate thought tokens, enabling improved reasoning with longer generations (Zelikman et al., 2024; Goyal et al., 2023). However, while this can boost performance, it incurs additional computational costs due to the need to decode (potentially many) reasoning tokens. LLM decoding can also be framed as sampling from a tree of possible sequences, inspiring research that uses inference-time compute to search through this tree (Yao et al., 2024; Liu et al., 2024; Han et al., 2024).

**Adaptive Computation Time in NN.** Several works explored the idea of learning how to adapt inference-time compute in neural networks (Graves, 2016; Dehghani et al., 2019; Banino et al., 2021). These works focus on networks with recurring components, where the decision is the number of times to pass the input through these components. Our work, however, is architecture-agnostic and focuses on adaptively selecting a decoding procedure. A recent line of works has demonstrated the compute-optimal inference can outperform simply using larger models (Snell et al., 2024; Wu et al., 2024). Here, we show how to realize these improvements in practice using learned difficulty predictors. Concurrent to our work, Manvi et al. (2024) demonstrated that language models can self-assess their response quality, enabling adaptive compute allocation.

## 6   CONCLUSION

We have introduced an approach for adaptively scaling test-time computation by predicting which queries would benefit the most from additional computation. We first showed that it is possible to learn lightweight models that predict marginal rewards on top of pre-trained LMs—indicating that LMs encode usable information about the reward distribution of their responses to inputs. We then presented an allocation algorithm that adaptively allocates computation to queries. Results in programming, mathematics, and chat show our approach gives significant reductions in computation at a target level of output quality, or improvements in quality given a fixed computation budget.

**Limitations:** In this work, we assumed that we have a training dataset of reasonable size ($> 3\text{K}$ queries) to train our marginal reward probe. We also assumed access to a verifier in the Math experiments. However, such verifiers will generally not be available, and using specialized reward models might be the more practical implementation of our method.

**Future Work:** The gap between our performance and the oracle indicates potential for improvement in marginal reward prediction. This could be addressed by exploring more advanced prediction models or by allocating additional inference-time computation for better marginal reward estimation.

ACKNOWLEDGMENTS

This work was supported by the MIT-IBM Watson AI Lab and the National Science Foundation and Intel under grants IIS-2212310 and CCF-2217064. Jacob Andreas is supported by a Sloan fellowship; Andi Peng is supported by the NSF GRFP and Open Philanthropy; and Idan Shenfeld is supported by the Qualcomm Innovation Fellowship.

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

## A  EXPERIMENT DETAILS

### A.1  MATH

**Dataset:** We use a subset of the Numina-COT Dataset (Li et al., 2024),which contains math problems from various sources. We filtered out multiple choice questions from the dataset. Our final dataset contained 15K training samples, 2K validation samples and 3K test samples.

**Model:** We use Mistral AI's **Mathstral-7B** model, which is specialized for mathematical tasks, and is based on Mistral 7B (Jiang et al., 2023). We use this model directly off-the-shelf and do not perform any fine-tuning.

**Verification Pipeline:** We used a 2 stage pipeline to verify the correctness of an answer. The first stage used the evaluation framework of Hendrycks et al. (2021). This framework expects the LM to enclose its answer within *boxed* {} and then runs an automated evaluation of the extracted answer against the ground truth answer. However, we found that the automated evaluation had a high false negative rate for various reasons:

1. **Answer not enclosed within *boxed* {}:** For some queries, the model had the right answer but did not enclose it inside *boxed* {}. This led to the responses being classified as incorrect.

2. **Errors in Checking:** We found that due to errors in checking, many correct responses were marked incorrect. Some examples of false negatives are:

   1. **Model Response:** frac{25}{2} ; **Ground Truth Answer**: 12.5

   2. **Model Response:** x=2 ; **Ground Truth Answer**: 2

   3. **Model Response:** -0.5 ; **Ground Truth Answer**: -frac{1}{2}

   4. **Model Response:** 11.00 ; **Ground Truth Answer**: 11

These errors were particularly harmful for the training of our marginal reward predictors. because queries predicted to be (and empirically) easy were being marked incorrect. Furthermore, our online allocation exacerbated this issue, by assigning very few samples to such queries.

To address this issue, we introduced a second stage in the verification pipeline that used a LM as an evaluator. Because stage 1 does not have false positives, we only used stage 2 for responses that were marked incorrect in stage 1. We used Llama-3.1-8B-Instruct with temperature set to 0.1 as our evaluation LM, and prompted it as follows:

> *You are a math evaluation agent. You are tasked with evaluating if the final answer from an excerpt of the response matches the given gold truth answer. The format or units of the response and gold truth answer might be different. However, you must evaluate if the answers are numerically equivalent/identical. Be extra careful when evaluating fractions, they must simplify to the same value Your response should be a single word followed by an explanation. 'YES' if the answers are equivalent and 'NO' if they are not.*
> *Examples:*
> *A) 7% and 7 are equivalent*
> *B) frac{10}{2} and frac {20}{4} are equivalent.*
> *C) 3,5,7 and 3,8,9 are not equivalent.*
>
> *Ground Truth Answer: <   >*
> *Response: <   >*

Note that if the model's response had an answer enclosed inside *boxed* {}, we only provided that extracted answer. If this was not the case, then we provided the model's entire response to the LLM evaluation agent.

**The 2-stage pipeline helped us significantly improve the evaluation process, as well as stabilize the training process for our marginal reward predictors. However, this pipeline is not without flaws, as we did notice examples a small number of false positives in the second stage.**

**Training:** We generate 128 responses for every query (temperature=0.7) and label them using the verification pipeline. We use the labels to compute the empirical mean success probabilities $\lambda_i$. The estimated $\lambda_i$ are then used as targets to train $\hat{\boldsymbol{\Delta}}(x; \theta)$, the marginal reward predictor.

**Evaluation:** Our adaptive allocation algorithms are assigned a maximum budget of $B_{max} = 128$ samples per query (that is the allocation for any query is capped at 128 samples).

## A.2 CODE

**Dataset:** We use a subset of TACO, a dataset focused on algorithmic code generation with problems sourced from various programming contest platforms such as CodeChef, CodeForces and Hacker-Rank. Our primary reason for selecting TACO was the availability of test cases for most problems, which is required to train a predictor of success probability. We filtered out problems that did not have a single unit test. We also filtered out problems that were sourced from *geeksforgeeks* and *aizu*, as the official evaluation framework did not support those. Our final dataset consisted of 10K training samples, 1K validation samples and 1K test samples. Note that these three splits in our experiments were extracted from official TACO *training set*. This is because the TACO test set has a very different distribution of problem difficulties, while our method aims to produce accurate *in-distribution* allocation of computation. We believe producing distributionally robust predictors is an important topic for future work.

**Model:** We used the **Starcoder-15B** model, which was finetuned and open-sourced by the authors of Taco.

**Training:** We generate 100 responses for every query (temperature=0.7) and label them using the unit-test verifier to obtain the mean success probability $\lambda_i$. The estimated $\lambda_i$ are then used as targets to train $\hat{\boldsymbol{\Delta}}(x; \theta)$, the marginal reward predictor.

**Evaluation:** We used the official evaluation framework released by the creators of the TACO dataset. A response was considered a success if it passed all available test cases. Our adaptive allocation algorithms are assigned a maximum budget of $B_{max} = 100$ samples per query (that is the allocation for any query was capped at 100 samples).

## A.3 CHAT

**Dataset:** We use a subset of the LMSYS-Chat dataset, which contains one million real-world conversations with 25 LLMs. We filter the dataset to only select samples in English and with less than 10 turns. We also filter out samples which were labelled redacted, as these were often artificially modified. Our final dataset consisted of 50K training samples and 5K test samples.

**Tranches Variant:** The tranches experiment uses a subset of $20\%$ from the full test set. To create this subset, we first generated multiple responses for each query in the test set and labeled them using the reward model. We then calculated the reward variance for each query and selected only those queries that fall in the lowest $10\%$ or highest $10\%$ of variance. In essence, the tranches test set is composed of queries with the lowest and highest variance in rewards, representing the two extremes. The main goal of the **tranches** experiment is to evaluate our method when the distribution of queries differs from typical datasets, which are often collected in somewhat controlled settings. Such datasets may not fully capture the true diversity of user queries that are encountered by general chatbots. While understanding the true distribution of user queries is beyond the scope of this work, the tranches experiment simulates a more extreme distribution to provide performance insights.

**Model:** We used the **Gemma-2B-it** model. We did not perform any fine-tuning.

**Training:** We generate 8 responses (temperature=0.7) for every query and label them using **NCSOFT/Llama-3-OffsetBias-RM-8B**, which was ranked amongst the top 10 on RewardBench at the time of writing. We then use bootstrapping to approximate $\boldsymbol{\Delta}_i$. The approximated $\boldsymbol{\Delta}_i$ are then used as targets to train $\hat{\boldsymbol{\Delta}}(x; \theta)$, the marginal reward predictor.

**Evaluation:** We evaluate using the same reward model as above. Our adaptive allocation algorithm was assigned a maximum budget of $B_{max} = 8$ samples per query.

### A.4 ROUTING: MODEL SIZE

**Dataset:** We used the same LLMSYS-Chat dataset that we used for the best-of-$k$ chat experiments.

**Model:** Our weak model was **Gemma-2B-it**. Our strong model was **Gemma-7B-it**. We did not perform any fine-tuning on either of these models.

**Training:** We generate 8 responses (at a temperature of 0.7) for every query with both the models. We labelled them using **NCSOFT/Llama-3-OffsetBias-RM-8B** as the reward model. Supervision for the reward model was then computed using a Monte Carlo estimate of:

$$(p^S \succ p^W | x) = \mathbb{E}_{y_1 \sim p^S, y_2 \sim p^W} [\sigma(r(x, y_1) - r(x, y_2))] \tag{11}$$

**Evaluation:** The adaptive allocation procedure uses the predictions of the marginal reward predictor to route the top $B^{th}$ percentile of queries to $p^S$. We evaluated using the same reward model mentioned above.

### A.5 ROUTING: VAS

**Dataset:** We use the harmless subset of the popular Anthropic-HH dataset. We sampled 8K samples randomly from the train set, and 400 samples from the test set.

**Model:** Our weak decoding procedure was a fine-tuned **Llama-7B** model. Our strong decoding procedure used an additional value function that was also based on the **Llama-7B** model.

**Training:** We generated 4 responses per query. The preference probability was then computed using a Monte Carlo estimate of:

$$p(p^S \succ p^W | x) = \mathbb{E}_{y_1 \sim p^S, y_2 \sim p^W} [\sigma(r(x, y_1) - r(x, y_2))] \tag{12}$$

We labelled the responses using **OpenAssistant/reward-model-deberta-v3-large-v2** as the reward model.

**Evaluation:** Our adaptive allocation used the predictions of the marginal reward predictor to route the top $B^{th}$ percentile of queries to $p^S$. We evaluated using the same reward model mentioned above.

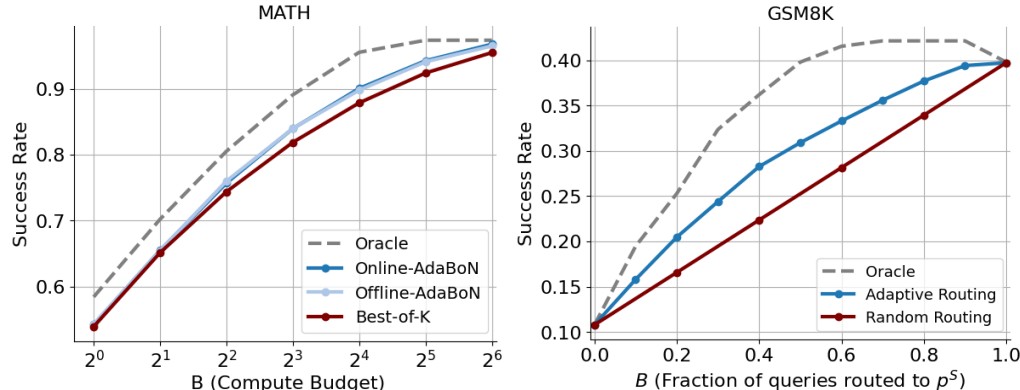

Figure 7: **Results on MATH and GSM8K**. We evaluate the MATH dataset using our adaptive best-of-$k$ approach and the GSM8K dataset using our routing approach. The results show that adaptive compute allocation improves performance on both benchmarks, demonstrating the ability of our learned predictor to effectively assess difficulty and the advantages of allocating compute adaptively.

# B    ADDITIONAL BENCHMARK RESULTS

## B.1    SETUP

**MATH**    We implement our adaptive best-of-k pipeline on the popular MATH benchmark (Hendrycks et al., 2021). Similar to our Numina Math experiments, we use Mistral AI's **Mathstral-7B** model. We use this model directly off-the-shelf and do not perform any fine-tuning. To train our probe, we generate $64$ responses per example for a total of $7,500$ training examples from the MATH dataset. We employed the 2-stage verifier pipeline used in the Numina experiments, described in Appendix A.1. We used the LoRA variant to learn the marginal rewards.

**GSM8K**    We implement our routing setting on the popular GSM8K dataset (Cobbe et al., 2021). We use the instruction-tuned **Gemma-2b-it** and **Gemma-7b-it** models as $p^W$ and $p^S$. We use these models directly off-the-shelf and do not perform any fine-tuning. We use a simple binary reward structure, where a correct answer is given a reward of $1$. To train our probe, we generate $16$ responses per example for a total of $7,500$ training examples from the GSM8K dataset. We perform LoRA fine-tuning of $p^W$ to learn the probe. As all problems in GSM8K have numerical answers, so we simply extract the last 2 numbers in the model's answer and classify an answer as correct if either of them match the ground truth answer.

## B.2    RESULTS

Figure 7 illustrates the results for both benchmarks, highlighting the effectiveness of adaptive compute allocation in both scenarios. In GSM8K, adaptive routing improves absolute success rates by up to $5\%$ compared to non-adaptive approaches while using same amount of compute. Similarly, in MATH, while the performance gains are more modest, adaptive compute allocation achieves comparable success rates while reducing compute usage by up to $2\times$ compared to the baseline best-of-$k$ approach. This demonstrates that our learned predictor effectively captures query difficulty and facilitates efficient compute allocation.

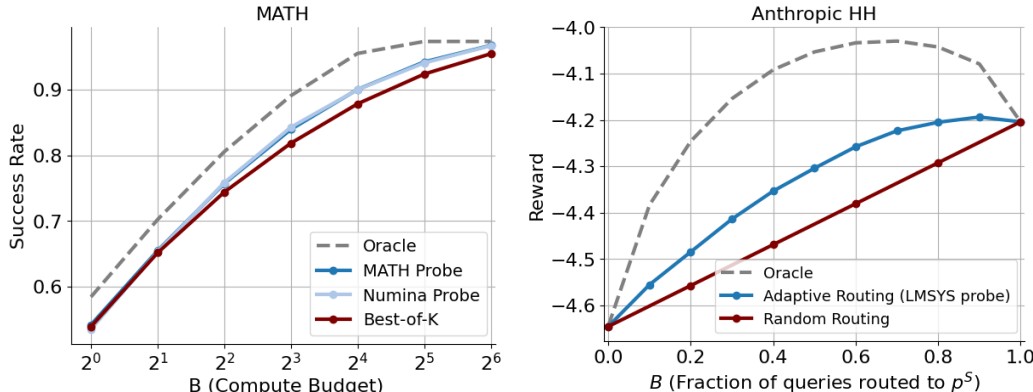

Figure 8: **Generalization results on MATH and Anthropic HH**. We evaluate the generalization of our probes trained on Numina and LMSYS by applying them to MATH and HH, respectively. The results demonstrate that our probes maintain high effectiveness for adaptive compute allocation, even when used outside their training data distribution. Remarkably, the Numina probe closely matches the performance of the MATH probe, suggesting that it is able to capture general features that are applicable across different mathematical datasets.

## C  GENERALIZATION TO DIFFERENT DATA DISTRIBUTIONS

In this section, we evaluate the effectiveness of our learned marginal reward predictors (probes) on adaptively allocating compute on queries outside their training distribution. Specifically, we conduct two generalization experiments: a) applying the probe trained on Numina dataset to the MATH dataset and b) applying the probe trained on LMSYS to Anthropic HH.

### C.1  SETUP

**Numina → MATH**   We use our best-of-$k$ probe, which was trained on the Numina dataset, and apply it to the MATH dataset. Although the original Numina dataset includes MATH as a subset, we excluded all examples sourced from MATH for this experiment.

**LMSYS → Anthropic HH**   We use our routing probe, which was trained on the LMSYS dataset and apply it to the Anthropic HH dataset (Bai et al., 2022). While LMSYS is a dataset containing real-world user conversations collected in the wild, data in HH was collected by instructing vetted master-qualified US-based MTurk workers with specific instructions. Thus, these datasets have significant differences in their user composition. We use the probe's predictions to route queries between **Gemma-2b-it** ($p^W$) and **Gemma-7b-it** ($p^S$). Thus, while the same decoding procedures are used for both datasets, the probe we use has only been trained on the LMSYS dataset.

### C.2  RESULTS

Figure 8 presents the results for our dataset generalization experiments. Our probes remain effective for adaptive compute allocation even when applied to data outside their training distribution, demonstrating strong generalization capabilities. Notably, the Numina probe matches the performance of the probe trained on MATH, suggesting that it successfully extracts general features related to mathematical problem difficulty from the language model's representations, independent of the dataset.

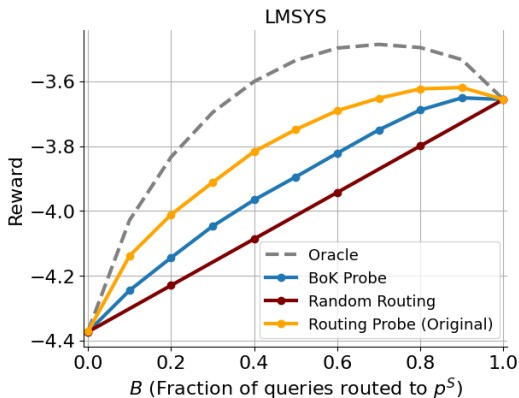

Figure 9: **Results of applying best-of-$k$ probe to Routing**. We use the probe trained for the best of-$k$ decoding method and apply it to routing. The results indicate that while there is some reduction in performance compared to a probe specifically trained for routing, the best-of-$k$ probe demonstrates effective generalization and is still able to deliver substantial gains.

## D    GENERALIZATION TO DIFFERENT DECODING METHODS

Our marginal reward predictors are typically trained for a specific decoding procedure. In this section, we explore whether these predictors can generalize to decoding procedures they were not explicitly trained on. Specifically, we conduct two generalization experiments: (a) evaluating a probe trained on the best-of-k approach when applied to routing, and (b) assessing the generalization of a probe trained on a specific decoding temperature on MATH across different decoding temperatures.

### D.1    MOTIVATION

**1. Applying BoK probe to Routing**    Through this experiment, we aim to understand if our adaptive best-of-$k$ (BoK) probe can be applied to routing. Intuitively, we want to extract a proxy of query difficulty from our BoK probe and use that for routing. In the best-of-$k$ setting, queries that have higher difficulty benefit from additional search (more samples). The difficult queries should thus have a high difference between the expected reward of best-of-k and the reward of taking a single sample. This range is given by:

$$\text{Difficulty}(x) \approx \mathbb{E}_{y \sim BoK}[r(x, y)] - \mathbb{E}_{y \sim p(\cdot|x)}[r(x, y)] \tag{13}$$

Using this, we can now approximate the preference probability (similar to Eqn. 8) of using the stronger decoding procedure $p^S$ as:

$$p(p^S \succ p^W | x) \approx \sigma(\text{Difficulty}(x)) \tag{14}$$

**2. Generalization across temperatures**    The targets for training our probes are generally obtained by using decoding with a fixed temperature (generally $0.7$). In this experiment, we want to test the generalization of our probe and the effectiveness of adaptive compute allocation if different decoding temperatures are used.

### D.2    SETUP

**1. Applying BoK probe to Routing**    We use the probe trained for the adaptive best-of-k experiments on LMSYS, (Section 4.1) and apply it routing on LMSYS. We use the instruction-tuned **Gemma-2b-it** and **Gemma-7b-it** models as $p^W$ and $p^S$. We compare performance against the probe that was trained specifically for routing in Section 4.2.

**2. Generalization across temperatures**    We evaluate the performance of our probe on the MATH dataset, focusing on its adaptability across different temperature settings. The probe is trained at

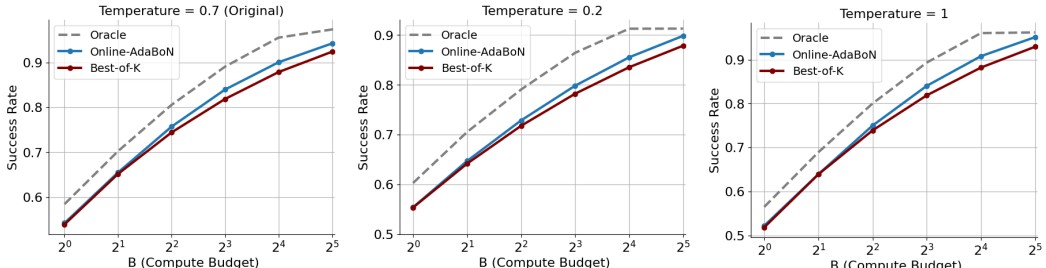

Figure 10: **Results on decoding with different temperatures**. We assess the performance of our probe, trained at a decoding temperature of 0.7, across various decoding temperatures. Despite being trained for a specific temperature, the probe remains effective across varying decoding temperatures. Intuitively, query difficulty is an intrinsic property and should be independent (or only weakly dependent) on the decoding temperature.

a temperature of 0.7, but we evaluate its generalization on two temperatures: 0.2 and 1. For each temperature, we perform decoding to estimate the empirical ground truth success probabilities and then test performance effectiveness of adaptive compute allocation. Since the scaling curves for Oracle and Best-of-$K$ are temperature-sensitive, we present three separate plots to illustrate the results for each temperature.

### D.3  RESULTS

Figures 9 and 10 present the results for the BoK-routing and temperature experiments respectively. We find that our BoK probe applied to routing continues to deliver significant gains compared to non-adaptive baselines. However, we do observe a drop in performance when compared to the probe that has been trained specifically for routing, indicating the existence of some generalization gap. In the temperature generalization experiments, we find that our probe remains effective across multiple decoding temperatures. These results suggest that our probes effectively capture a notion of query difficulty that shows some degree of consistency for different decoding procedures i.e., a query considered difficult for procedure A is also likely to be considered difficult for procedure B.

