# OpenReview forum: "Learning How Hard to Think: Input-Adaptive Allocation of LM Computation"
_ICLR.cc/2025/Conference — ICLR 2025 Poster_

### Official Review · Reviewer_amKi · 2024-10-27

**Soundness:** 3
**Presentation:** 3
**Contribution:** 2
**Rating:** 6
**Confidence:** 4

**Summary:**

This paper proposes a method for the adaptive allocation of decoding computation. By employing an LLM-based probe to predict the difficulty of a given query, the approach dynamically adjusts the allocation of decoding resources. The authors validate the method’s effectiveness across coding, math, and chat tasks. Results demonstrate that, under computational constraints, this approach outperforms the baseline BoK method.

**Strengths:**

1. This paper achieves efficient decoding from a different perspective, showing clear improvements over the original BoK method.
2. The authors apply their method across three distinct domains—code, math, and chat—demonstrating generalizability of their method.

**Weaknesses:**

1. In the experiments for code and math, the authors employ less-used benchmarks rather than widely adopted ones like HumanEval and MATH, raising concerns about the method’s applicability to broader tasks.

2. The paper's baseline comparison is limited to the BoK method, lacking comparative experiments with other stronger efficient decoding methods, such as Speculative Decoding.

**Questions:**

1. Please explain the choice of benchmarks and how they compare to HumanEval and MATH in terms of difficulty distribution. Or please add more benchmarks like HumanEval and MATH.

2. If a more complex decoding method, such as MCTS, is employed, would it necessitate retraining the probe? This could suggest a mismatch between the model's capabilities when using more advanced decoding methods and the probe's predictions. Additionally, it raises the question of whether the probe's prediction accuracy may be affected by factors such as varying prompts or decoding methods, and whether the probe demonstrates robustness under these conditions. Please explain the discuss the generalizability of the probe.

---

> ### Author Response · Authors · 2024-11-19
> **Response to Reviewer amKi (Part 1)**
>
> **Q1. Please explain the choice of benchmarks and how they compare to HumanEval and MATH in terms of difficulty distribution.**
>
> **A1.** Thanks for the question! To address concerns about generalization to other datasets, we have added a new set of  benchmark results (**Appendix B** in the revised submission) on MATH and GSM8K. Specifically, we evaluate on MATH using our adaptive best-of-k approach and on GSM8K with our routing approach. We find that adaptive compute allocation improves performance on both benchmarks. Notably, adaptive routing on GSM8K improves absolute success rates by up to 5% (a relative increase of nearly 20%) while using the same amount of compute as non-adaptive methods.
>
> We also present results on Anthropic HH [3] (**Appendix C**), which has been used as a standard benchmark in RLHF. However, HumanEval/MBPP do not have training datasets and thus we are unable to train difficulty models for these benchmarks.
>
> **Q2. The paper's baseline comparison is limited to the BoK method, lacking comparative experiments with other stronger efficient decoding methods, such as Speculative Decoding.**
>
> **A2.** Thank you for your question. We would like to emphasize that the contribution of this paper  is  not a particular decoding method (such as adaptive best-of-k or routing), but instead to show that adaptive test-time compute allocation can be beneficial across a diverse set of existing decoding methods. In this sense, the “baseline” methods are those that allocate a fixed amount of computation per query. Concurrent work by Snell et al.[2], also shows the value of adaptive computation and does not consider method-specific baselines.
>
> Most of the relevant work in the area presents different methods to use test-time computation (chain-of-thought, generate and revise, MCTS, etc) . In this work, we consider a different axis which tries to adaptively allocate this computation, making our method complementary to most test-time methods.
>
> Speculative decoding is only applicable to our routing setting with LM size (which is only 1 out of our 5 experiments). Moreover, speculative decoding is not input-adaptive (our main novelty) and uses a fixed frequency to query the large model, making it combinable with our method. Specifically, speculative decoding uses a fixed, query-invariant frequency to query the larger model. Combining with our framework would imply input-adaptively choosing at what frequency queries should be verified on the larger model.

---

> > ### Author Response · Authors · 2024-11-19
> > **Response to Reviewer amKi (Part 2)**
> >
> > **Q3.  If a more complex decoding method, such as MCTS, is employed, would it necessitate retraining the probe? This could suggest a mismatch between the model's capabilities when using more advanced decoding methods and the probe's predictions. Additionally, it raises the question of whether the probe's prediction accuracy may be affected by factors such as varying prompts or decoding methods, and whether the probe demonstrates robustness under these conditions. Please explain the generalizability of the probe.**
> >
> > **A3.** Thank you for the great question! Our probes are trained specific to a decoding procedure but have a good degree of generalization. We run 4 experiments to evaluate the generalization of our probes:
> >
> > - **Generalization to Different Data Distributions (Appendix C)**: These experiments evaluate the probes on queries outside their training distribution. The chat datasets we consider (see point 2 below) naturally have varying prompts/ prompting styles as they were collected in significantly different ways.
> >     1. **Applying the difficulty model trained on the Numina dataset to the popular MATH benchmark (Figure 8)**: We find that our probe matches the performance of a probe trained on MATH! **This suggests that our difficulty model is able to capture general features that are applicable across different mathematical datasets.**
> >
> >   2.  **Applying the difficulty model trained on the LMSYS dataset to the popular Anthropic HH dataset (Figure 8)**: LMSYS and Anthropic HH are both chat datasets but were collected in significantly different ways [1,3]. Despite this, we find that our difficulty model generalizes well and using it to route queries adaptively is significantly better than non-adaptive methods. In particular, we can achieve up to a 40% reduction in calls to the more expensive decoding scheme while maintaining similar levels of reward.
> >
> > - **Generalization to Different Decoding Procedures (Appendix D)**: These experiments evaluate the probes on decoding procedures that they were not trained for.
> >      1. **Applying our best-of-k probe to routing (Figure 9)**: We use the probe trained for the best of-k decoding method and apply it to routing. The results indicate that while there is some reduction in performance compared to a probe specifically trained for routing, the best-of-$k$ probe demonstrates effective generalization and is still able to deliver substantial gains compared to random routing.
> >
> >      2. **Generalization across temperatures (Figure 10)**: We assess the performance of our probe, trained at a decoding temperature of 0.7, across various decoding temperatures. Despite being trained for a specific temperature, the probe remains effective across varying decoding temperatures.
> >
> >
> > Intuitively, while stronger decoding procedures might find queries less difficult, the relative difficulty of queries might be consistent across different decoding methods. Thus, even if our probes lose calibration on difficulty, if they are able to preserve relative difficulty, they can still be effective for downstream compute allocation.
> >
> > We do acknowledge that if a decoding procedure is vastly different from what the probe is trained on, some performance degradation is likely. Here, we would like to note that the probes we train are extremely lightweight and the entire pipeline can be run in less than 12 hours. Thus, training a new probe when switching to a significantly different decoding procedure should not add a lot of overhead. Finally, it might also be possible to train multi-task probes which are conditioned on the decoding procedure itself, although we leave this for future work.
> >
> > &nbsp;
> >
> > [1]: Zheng, Lianmin, et al. "Lmsys-chat-1m: A large-scale real-world llm conversation dataset." arXiv preprint arXiv:2309.11998 (2023).
> >
> > [2]: Bai, Yuntao, et al. "Training a helpful and harmless assistant with reinforcement learning from human feedback." arXiv preprint arXiv:2204.05862 (2022).
> >
> > [3]: Snell, Charlie, et al. "Scaling llm test-time compute optimally can be more effective than scaling model parameters." arXiv preprint arXiv:2408.03314 (2024).

---

> > > ### Comment · Reviewer_amKi · 2024-11-20
> > >
> > > Thank you for your response. You have conducted very detailed analysis and experiments, which resolved my questions. I will increase the score. Thank you.

---

> > > > ### Author Response · Authors · 2024-11-20
> > > > **Response to Reviewer amKi**
> > > >
> > > > Thank you for your feedback, which has contributed greatly to improving this paper! Are there any other changes we can make that would allow you to increase your score further?

---

### Official Review · Reviewer_igGM · 2024-11-02

**Soundness:** 2
**Presentation:** 3
**Contribution:** 3
**Rating:** 6
**Confidence:** 4

**Summary:**

This work proposes an input-adaptive computation allocation mechanism for improving the efficiency of test-time computation. The core idea is to train a model that predicts the distribution of rewards given a query and a budget. It incorporates training an MLP LM head and LoRA as the reward predictor that estimates the difficulty of a batch of queries. The proposed adaptive best-of-k outperforms the efficiency of standard best-of-k baselines in math, code, and chat domains. In addition, the author demonstrates the improvement in routing in terms of different model sizes and decoding schemes. The additional case study in inspecting the allocation of computation at different budgets is intriguing.

**Strengths:**

1. Scaling the test-time compute is effective but costly, this work contributes to a timely direction with a smart input-adaptive allocation scheme improving test-time efficiency.

2. The empirical improvement in efficiency is noticeable, and this work has covered adaptive allocation in representative popular subdomains: sampling, model size, and decoding method.

3. The presented analysis in Figure 6 is intuitive.

**Weaknesses:**

1. The selection of datasets and backbone language models may be questionable. I suspect this method should be ideally generalizable across tasks, however, only a single data in each domain is selected. I expect to see more tasks like HumanEval, MBPP for coding, Hendrycks MATH, and GSM for math. Meanwhile, for each domain, the author selects a specific backbone LM rather than the same choice across all tasks. This may raise concerns about the generalization of the proposed method.

2. The underlying difficulty of this method is to actually train a very good difficulty estimator. However, the training difficulty, and the heavy training data resource requirement for learning a good reward predictor have not been explicitly discussed in the context. Moreover, it is highly dependent on the task, and I suspect the difficulty of some tasks will not be easy to predict.

3. The proposed method only considers the query for training the reward predictor. However, though there is a latency for querying the model, I suspect introducing $y$ will be more informative to reflect the difficulty of a task.

4. In Figure 3 (middle), besides the left bottom and right top clusters, the rest correlation appears to be relatively poor. Therefore, I suspect the efficiency gain could be mostly coming from predicting “unanswerable” for the queries in the left bottom regions and putting 0 costs there, also assigning a minimum budget to always correct questions. However, the middle region is actually the region that should benefit from a smart computation allocation scheme, and the correlation is not convincing here.

**Questions:**

1. Though I understand using a query only to predict the reward should incur less latency, will $y$ be more informative and easier to train the predictor?

2. Could you please report the Spearman Correlation in Figure 3 (b, middle column)?

3. Could you provide more clarification on the computing budget? Is it based on the inference calls?

I will be happy to raise my score if the author could address the aforementioned limitations and concerns.

---

> ### Author Response · Authors · 2024-11-19
> **Response to Reviewer igGM (Part 1)**
>
> **Q1. I suspect this method should be ideally generalizable across tasks, however, only a single data in each domain is selected. I expect to see more tasks like HumanEval, MBPP for coding, Hendrycks MATH, and GSM for math.**
>
> **A1.** Thank you for the suggestion. To address this concern, we have added a new set of  benchmark results (**Appendix B** in the revised submission) on MATH and GSM8K. Specifically, we evaluate on MATH using our adaptive best-of-k approach and on GSM8K with our routing approach. We find that adaptive compute allocation improves performance on both benchmarks (**Figure 7**). Notably, adaptive routing on GSM8K improves absolute success rates by up to 5% (a relative increase of nearly 20%) while using the same amount of compute as non-adaptive methods. We also present results on Anthropic HH [1] (**Figure 8**), which has been used as a standard benchmark in RLHF. However, HumanEval/MBPP do not have training datasets and thus we are unable to train difficulty models for these benchmarks.
>
> &nbsp;
>
> **Q2. The author selects a specific backbone LM rather than the same choice across all tasks. This may raise concerns about the generalization of the proposed method.**
>
> **A2.** Thank you for your question! Although the downstream compute allocation is independent of the base LLM, our difficulty model is learned on top of the base LLM’s representations. Our main reason for using different LLMs was to show that we can learn effective difficulty predictors across a variety of base LLM models.
>
> We also provide new results on GSM8K with the Gemma family of models (**Appendix B**) and find that the performance gains are significant and follow the trend we observed for specialized Math models. Thus, all of our chat experiments (with the exception of value-augmented search for which no Gemma model is openly available) and the new GSM8K experiment use the Gemma family of models.
>
> &nbsp;
>
> **Q3. The heavy training data resource requirement for learning a good reward predictor have not been explicitly discussed in the context.**
>
> **A3.** Thank you for pointing this out! The probes we train are actually very lightweight and even when we use LoRA, the average training time is 3-4 hours on 1 A100 GPU. Constructing the training dataset requires MC sampling to estimate the targets and using VLLM, we found that this generally took anywhere between 4 hours (Math, 10K examples) to 10 hours (LMSYS, 50K examples). We also acknowledge that we need a training dataset for our probe, and have added this to the limitations section. Let us know if there is anything else we can do to address this.
>
> &nbsp;
>
>  **Q4. The underlying difficulty of this method is to actually train a very good difficulty estimator. I suspect the difficulty of some tasks will not be easy to predict. However, though there is a latency for querying the model, I suspect introducing y will be more informative to reflect the difficulty of a task.**
>
> **A4.** While perfectly predicting difficulty is indeed a challenging problem, our experiments show that even somewhat noisy predictions are good enough:  on all 3 domains (and our new benchmark results),we are still able to predict difficulty to a degree that makes it useful for the downstream application of adaptive compute allocation.
>
> Developing better difficulty models, although beyond the scope of this work, is definitely worth exploring and can significantly boost performance. We fully agree with the reviewer’s comment that conditioning on the response y can boost the performance of the difficulty model and are actually considering this for future work! We are considering a setting where we allocate some amount of initial computation to each query and use the evaluation of those responses to refine our difficulty estimate. In such a setting, conditioning on the response y can significantly improve performance. However, this will also substantially increase latency and comes with other computational costs.

---

> > ### Author Response · Authors · 2024-11-19
> > **Response to Reviewer igGM (Part 2)**
> >
> > **Q5. In Figure 3 (middle), besides the left bottom and right top clusters, the rest correlation appears to be relatively poor. Therefore, I suspect the efficiency gain could be mostly coming from predicting “unanswerable” for the queries in the left bottom regions and putting 0 costs there, also assigning a minimum budget to always correct questions. Could you please report the Spearman Correlation in Figure 3 (b, middle column)?**
> >
> > **A5.** The Spearman Correlation for Code (**Fig 3, top row**)  is **0.79**  and for Numina (**Fig 3, bottom row**) is **0.8**.
> >
> > There is indeed more signal in the extreme queries but the predictions in the moderate difficulty ranges are also well correlated. Intuitively, this makes sense as it might be easier for the model to predict if it knows/does not know something, but harder to predict its distribution over different answers. Moreover, while there might be more variance for predictions in the moderate regions, we find that our difficulty predictors are actually well-calibrated.
> >
> > Also, note that ~90% queries on Numina can actually be solved with sampling. Thus, although it appears that there is high density in the lower extremes, assigning 0 cost to these predictions is actually very suboptimal when given large budgets. Thus, even for extreme values, it is important to have well-calibrated estimation.
> >
> > &nbsp;
> >
> > **Q6. Could you provide more clarification on the computing budget? Is it based on the inference calls?**
> >
> > **A6.** The framework we present actually allows computing budget to be defined in different ways such as inference calls, tokens, length, etc. For our specific experiments:
> >
> > - **Adaptive Best-of-K (Lines 256-259)** : Here, compute budget refers to the number of responses (inference calls) to sample for each query. As a simple example, consider we have 100 queries and a total budget of 1000 inference calls. Then the baseline best-of-k will take 10 responses for each query, while our method will decide this allocation adaptively.
> >
> > - **Routing**: Here, only 1 inference call is made per query but that call may be made to a strong decoding procedure or a weak decoding procedure. The total compute budget defines the fraction of calls that can be made to the strong decoding method. For example, B=0.7 implies that 70% of the queries should be routed to the strong procedure. We realized that we had not explicitly defined this in the paper and have added it (**Lines 428-429**).
> >
> > &nbsp;
> >
> > [1]: Bai, Yuntao, et al. "Training a helpful and harmless assistant with reinforcement learning from human feedback." arXiv preprint arXiv:2204.05862 (2022).

---

> ### Comment · Reviewer_igGM · 2024-11-19
> **Response to Author**
>
> Thank the author for the response. Since it has mostly addressed my concerns, I have increased my score.
>
> Regarding A5, could you please also report the correlation if you have excluded the extreme regions' points? I am curious to know the moderate region's correlation.

---

> > ### Author Response · Authors · 2024-11-20
> > **Response to Reviewer igGM**
> >
> > Thank you for engaging with the paper! To compute correlation in the moderate region, we select the data where the ground truth probabilities fall within the [0.1, 0.9] range and report the correlation on this subset:
> >
> > **Numina** (50% of the total dataset size): 0.61
> >
> > **Code** (30% of the total dataset size): 0.53
> >
> > As expected, there is a drop but predictions in the moderate difficulty ranges are also well correlated. This aligns with our intuition that it is easier for the model to predict if it knows/does not know something, but harder to predict its distribution over different answers.
> >
> > &nbsp;
> >
> > Thank you for your feedback, which has contributed greatly to improving this paper! Are there any other changes we can make that would allow you to increase your score further?

---

### Official Review · Reviewer_NnGM · 2024-11-04

**Soundness:** 3
**Presentation:** 3
**Contribution:** 3
**Rating:** 8
**Confidence:** 4

**Summary:**

Presents an input-adaptive method for test-time compute-allocation. Decoding methods apply either sequential (eg weak vs strong model) or parallel compute (eg more samples in best-of-n). For a given method, this paper proposes to predict the marginal utility of every unit of computation, then use these predictions to optimize compute allocation. The paper proposes to predict these utilities given only the input.

The resource allocation problem can be solved in an offline manner given a fully observed dataset, referred to as online allocation in the paper, or solved via online access to only a partially observed dataset, referred to as offline allocation in the paper.

Experimental results across coding, math, and chat indicate that utility prediction is difficult at the extremes, and allocation decisions are sensitive to utility errors. Overall, adaptive allocation outperforms uniform or random allocations. The partially-observed strategy empirically often does better than the fully-observed case, possibly due to coarsening effects that hide errors in utility prediction.

**Strengths:**

The paper tackles a novel and timely problem, and offers a reasonable approach. The paper is clearly written.

**Weaknesses:**

A small criticism is the naming convention of online versus offline. Online optimization refers to "optimization problems having no or incomplete knowledge of the future (online)," which is not how online is used in this paper.

Other than that, this paper is a good step in improving adaptive test-time compute, identifying the importance of accurate utility estimation in problems with very low success rates.

**Questions:**

Drawing inspiration from the online secretary problem, it would be interesting to see how online estimation of pass rates for coding can aid utility estimation. For example, one could increase the total computation budget and, for each problem, reserving some of that budget to utility estimation. This would alleviate some of the burden from the prediction model.

---

> ### Author Response · Authors · 2024-11-19
> **Response to Reviewer NnGM**
>
> **Q1. Drawing inspiration from the online secretary problem, it would be interesting to see how online estimation of pass rates for coding can aid utility estimation. For example, one could increase the total computation budget and, for each problem, reserving some of that budget to utility estimation.**
>
> **A1.** Thank you for your suggestion, this is an extremely interesting idea and one we had already starting thinking about as a follow-up study! In a serial setting (and unlike the standard secretary problem), individual decoding results y can provide fine-grained information about the distribution of future outputs from different sampling methods. A good difficulty model can condition on ys, and many more complex decoding strategies are possible. We think this is a great direction for future research.
>
> &nbsp;
>
> **Q2. Online vs offline**
>
> **A2.** Thanks for pointing this out—we will clarify in our final revision.

---

### Official Review · Reviewer_46zi · 2024-11-08

**Soundness:** 3
**Presentation:** 2
**Contribution:** 2
**Rating:** 6
**Confidence:** 3

**Summary:**

This paper presents an approach to adaptively allocate computational resources for language model (LM) decoding based on input difficulty. The authors propose a framework that predicts the marginal benefit of additional computation for each query, enabling dynamic adjustment of decoding procedures to maximize efficiency without sacrificing output quality. They demonstrate their method across tasks in math, code generation, and dialogue, achieving up to 50% reduction in compute usage in some cases. The paper also introduces two adaptive procedures—best-of-k sampling and routing between models of varying complexity—and provides a thorough evaluation using both online and offline allocation strategies.

**Strengths:**

- The paper adeptly formulates the "adaptive computation scaling allocation" in the context of LM decoding, addressing a topic that is both timely and relevant.
- The proposed computation-allocation framework is comprehensive, covering various cases and scenarios, including binary reward, pairwise optimization in routing, and both online and offline design considerations.
- The experiments conducted on three diverse and representative domains demonstrate the efficiency and efficacy of the proposed computation-allocation strategies.

**Weaknesses:**

- The main concern is that the current computation-allocation solution is only evaluated in scenarios with identical distributions (i.e., the training data used to train the difficulty model comes from the same distribution as the test set). It is unclear whether the trained difficulty model generalizes to other distributions. The generalizability of the difficulty model is crucial for determining the practicality of the proposed computation-allocation framework.
- Following from the above, since the choice of LLMs does not seem to affect the evaluation of the proposed method’s efficacy, why not select a single fixed LLM, such as Llama3-7b-Instruct? By doing so, it might be easier to assess the generalizability of the method. (Please correct me if there is an issue with my understanding.)
- The implementation of the baselines is weak, with only one effective but not particularly practical baseline (best-of-k and random) in each scenario. Between the proposed method and these baselines, there are likely other reasonable approaches that could better demonstrate the effectiveness of the proposed framework.
- The related work section is too concise and lacks comprehensiveness, especially in the discussion of relevant adaptive computing research. Only one recently published paper is mentioned, which undermines the paper's completeness and contextual grounding.

**Questions:**

- Typos:
    - Line 352: "which in an" should be "which is an".
    - "LoRa" should be changed to "LoRA".
    - In Figure 1, "Large LM" should be changed to "large LM" for consistency.
- What's the definition of "N" in equation (10)?

---

> ### Author Response · Authors · 2024-11-19
> **Response to Reviewer 46zi (Part 1)**
>
> **Q1. Does the computation allocation solution generalize to new data distributions?**
>
> **A1**.  Great question! To answer it, we ran two new experiments (see **Appendix C** in the revised submission) to evaluate how our difficulty model generalizes to data distributions it was not trained on:
>
> 1. **Applying the difficulty model trained on the Numina dataset to the popular MATH benchmark (Figure 8)**: We find that our difficulty model shows strong generalization and that downstream adaptive compute allocation with this probe leads to significant gains over non-adaptive baselines. Interestingly, we also find that our difficulty model matches the performance of a difficulty model trained on MATH. This suggests that our difficulty model is able to capture general features that are applicable across different mathematical datasets.
>
> 2. **Applying the difficulty model trained on the LMSYS dataset to the popular Anthropic HH dataset (Figure 8)**: LMSYS and Anthropic HH are both chat datasets but were collected in significantly different ways [1,2]. Despite this, we find that our difficulty model generalizes well and using it to route queries adaptively is significantly better than non-adaptive methods. In particular, we can achieve up to a 40% reduction in calls to the more expensive decoding scheme while maintaining similar levels of reward.
>
> In addition to these results, we would like to highlight that the LMSYS dataset is itself very diverse and captures a large distribution of users. In particular, the LMSYS dataset is composed of real-world user conversations collected from 210K unique IP addresses [1]. Although we do agree and note in the paper that this still maybe somewhat controlled (for example, the website used for collection may have users that are primarily LLM hobbyists), we believe that the chat results on LMSYS (for routing and adaptive BoK) also demonstrate some degree of generalizability of our difficulty model.
>
>  Finally, we also present results that show generalization of our difficulty model to decoding procedures that it was not trained for (see **Appendix D**).
>
> &nbsp;
>
>
> **Q2. Following from the above, since the choice of LLMs does not seem to affect the evaluation of the proposed method’s efficacy, why not select a single fixed LLM, such as Llama3-7b-Instruct?**
>
> **A2.** Thank you for your question! Although the downstream compute allocation is independent of the base LLM, our difficulty model is learned on top of the base LLM’s representations. Our main reason for using different LLMs was to show that we can learn effective difficulty predictors across a variety of base LLM models.
>
> We also provide new results on GSM8K with the Gemma family of models (see **Appendix B**) and find that the performance gains are significant and follow the trend we observed for specialized Math models. Thus, all of our chat experiments (with the exception of value-augmented search for which no Gemma model is openly available) and the new GSM8K experiment use the Gemma family of models.
>
> &nbsp;
>
> **Q3. The implementation of the baselines is weak, with only one effective but not particularly practical baseline (best-of-k and random) in each scenario. There are likely other reasonable approaches that could demonstrate the effectiveness of the proposed framework.**
>
> **A3.** Thank you for your question. We would like to emphasize that the contribution of this paper  is  not a particular decoding method (such as adaptive best-of-k or routing), but instead to show that adaptive test-time compute allocation can be beneficial across a diverse set of existing decoding methods. In this sense, the “baseline” methods are those that allocate a fixed amount of computation per query. Concurrent work by Snell et al.[3], also shows the value of adaptive computation and does not consider method-specific baselines.
>
> Finally, most of the relevant work in the area presents different methods to use test-time computation (chain-of-thought, generate and revise, MCTS, etc) . In this work, we consider a different axis which tries to adaptively allocate this computation, making our method complementary to most of these test-time methods.
>
> Please let us know if there are specific comparisons you would like to see!

---

> > ### Author Response · Authors · 2024-11-19
> > **Response to Reviewer 46zi (Part 2)**
> >
> > **Q4. The related work section is too concise and lacks comprehensiveness, especially in the discussion of relevant adaptive computing research. Only one recently published paper is mentioned, which undermines the paper's completeness and contextual grounding.**
> >
> > **A4.** Thank you for your suggestion. We have added 5 new papers to our related work, of which 3 are related to adaptive compute allocation. **However, please note that 2 of the adaptive compute papers are concurrent work, and were released publicly only after the ICLR submission deadline**. We would also be happy to consider any specific papers the reviewer believes we may have overlooked.
> >
> > -  Manvi, Rohin, Anikait Singh, and Stefano Ermon. "Adaptive Inference-Time Compute: LLMs Can Predict if They Can Do Better, Even Mid-Generation." arXiv preprint arXiv:2410.02725(2024).
> >
> > - Wu, Yangzhen, et al. "Inference Scaling Laws: An Empirical Analysis of Compute-Optimal Inference for Problem-Solving with Language Models." arXiv preprint arXiv:2408.00724(2024).
> >
> > - Zhang, Kexun, et al. "Scaling LLM Inference with Optimized Sample Compute Allocation." arXiv preprint arXiv:2410.22480(2024).
> >
> > - Zelikman, Eric, et al. "Quiet-star: Language models can teach themselves to think before speaking." arXiv preprint arXiv:2403.09629 (2024).
> >
> > - Goyal, Sachin, et al. "Think before you speak: Training language models with pause tokens." arXiv preprint arXiv:2310.02226 (2023).
> >
> > &nbsp;
> >
> > **Q5. What's the definition of "N" in equation (10)?**
> >
> > **A5.** The capitalization is a typo on our end and we have fixed it. It should be $n$, which is the number of queries in the set.
> >
> > &nbsp;
> >
> > **Regarding Typos**:
> >
> > Thank you for pointing these out, we have fixed all of them!
> >
> > &nbsp;
> >
> > [1]: Zheng, Lianmin, et al. "Lmsys-chat-1m: A large-scale real-world llm conversation dataset." arXiv preprint arXiv:2309.11998 (2023).
> >
> > [2]: Bai, Yuntao, et al. "Training a helpful and harmless assistant with reinforcement learning from human feedback." arXiv preprint arXiv:2204.05862 (2022).
> >
> > [3]: Snell, Charlie, et al. "Scaling llm test-time compute optimally can be more effective than scaling model parameters." arXiv preprint arXiv:2408.03314 (2024).

---

> > > ### Comment · Reviewer_46zi · 2024-11-21
> > >
> > > Thank you to the authors for their response. It answered most of my concerns and made the paper feel more complete. I have raised my score and am leaning toward accepting the paper.

---

> > > ### Public Comment · ~Xinglin_Wang1 · 2024-11-22
> > > **Suggestion on another highly relevant related work**
> > >
> > > Congratulations to the authors on their high-quality work! Here, I have listed another study that is highly relevant to this paper as a reference [1], hoping it could be helpful for the authors' related work section.
> > >
> > > [1] Make every penny count: Difficulty-adaptive self-consistency for cost-efficient reasoning 2024.8.24

---

> > > > ### Author Response · Authors · 2024-11-22
> > > >
> > > > Thank you for pointing out this relevant piece of concurrent work, we'll mention it in the final version of the paper!

---

### Author Response · Authors · 2024-11-19
**Global Comments and Revision Summary**

We thank the reviewers for their time and effort in reviewing this paper. We are happy that the reviewers found our work timely and comprehensive. The insightful comments received have helped us refine this paper. Below, we summarize the main concerns raised and outline the additional experiments conducted to address them:

1. **Evaluation on standard benchmarks**: We have included new benchmark results on MATH and GSM8K (**Appendix B** in the revised submission). These results confirm that adaptive compute allocation remains beneficial in these settings.


2. **Generalization of the Learned Difficulty Model to Unseen Data Distributions**: We conducted two additional experiments (**Appendix C**) to assess how well our difficulty model generalizes to data outside its training distribution. The results demonstrate strong generalization: adaptive compute allocation using our model consistently outperforms non-adaptive baselines.

3. **Generalization of the Learned Difficulty Model to New Decoding Procedures**: To evaluate robustness across decoding methods, we ran two new experiments (**Appendix D**) testing our difficulty model on decoding procedures it was not trained for. While there is a slight performance drop, the model still delivers strong results, indicating that our difficulty model has learned a general notion of query difficulty which is applicable to a range of decoding procedures.  its robustness to varied decoding approaches.


4. **Reasoning for using different LLMs for different experiments**: Since our difficulty model is learned on top of an LLM’s representations, our primary reason for using different LLMs was to demonstrate that it is possible to learn effective difficulty models for a variety of base LLM models. Since that raised concern among reviewers, we tried to address that (within the time limit of the rebuttal), and now, with the inclusion of our new results, all of our chat experiments (with one exception) and one Math experiment use the Gemma family of models.

---

### Meta-Review · Area_Chair_MNXE · 2024-12-19

**Metareview:**

The paper presents an adaptive computation allocation approach for LM decoding that predicts input difficulty to optimize resource usage. Key strengths include: comprehensive evaluation across diverse domains (code, math, chat), significant compute reduction (up to 50%) without quality loss, and strong generalization results on standard benchmarks (MATH, GSM8K). Main weaknesses are: initial limitation to same-distribution evaluation, use of different LLMs across experiments, and moderate prediction accuracy for medium-difficulty cases. However, authors adequately addressed these through additional experiments showing cross-dataset/model generalization. The paper makes a valuable contribution to efficient LM deployment and is recommended for acceptance.

**Additional Comments On Reviewer Discussion:**

Reviewers initially raised concerns about generalization across datasets/models and baseline comparisons. Authors addressed these through new experiments on MATH/GSM8K benchmarks showing consistent benefits, clarified the complementary nature of their method to existing approaches, and demonstrated probe generalization across distributions and decoding methods. All reviewers increased scores after rebuttal.

---

### Decision · Program_Chairs · 2025-01-22

Accept (Poster)